# Emergence of crucial evidence catalyzing the origin tracing of SARS-CoV-2

**Shunmei Chen**[1,2☯], **Cihan Ruan**[3☯], **Yutong Guo**[1], **Jia Chang**[1], **Haohao Yan**[1], **Liang Chen**[2], **Yongzhong Duan**[2], **Guangyou Duan**[4], **Jinlong Bei**[5], **Xin Li**[1]*, **Shan Gao**[1]*

**1** College of Life Sciences, Nankai University, Tianjin, Tianjin, P.R.China, **2** Biomedical Engineering Research Institute, Kunming Medical University, Kunming, Yunnan, P.R.China, **3** Department of Computer Science and Engineering, Santa Clara University, Sant Clara, California, United States of America, **4** School of Life Sciences, Qilu Normal University, Jinan, Shandong, P.R.China, **5** Agro-biological Gene Research Center, Guangdong Academy of Agricultural Sciences, Guangzhou, Guangdong, P.R.China

☯ These authors contributed equally to this work.

\* lix1980@nankai.edu.cn (XL); gao_shan@mail.nankai.edu.cn (SG)

**Data Availability Statement:** All relevant data are within the paper and its Supporting Information files.

**Funding:** The author(s) received no specific funding for this work.

## Abstract

Since the emergence of severe acute respiratory syndrome coronavirus-2 (SARS-CoV-2), its genetic and geographical origins remain unclear, resulting in suspicions about its natural origin. In one of our previous studies, we reported the presence of a furin cleavage site RRAR in the junction region between S1 and S2 subunits of the spike protein, which was discovered as the first crucial clue for the origin tracing of SARS-CoV-2. In the present study, we conducted an integrative analysis of new genome data from bat *Sarbecovirus* strains reported after the COVID-19 outbreak. The primary results included the identification of BANAL-20-52, Rp22DB159, and S18CXBatR24 as three close relatives of SARS-CoV-2 and the successful detection of seven out of nine key genomic features (designated as RC0-7 and *ORF8*) observed in wild types of SARS-CoV-2 in the three close relatives from Laos, Vietnam, and Yunnan province of China, respectively. The most significant contribution of the present study lies in the detection of RC1 in wild genotype in a bat *Sarbecovirus* population BANAL-20-52 belonging to. Encoding a segment of the *NSP3* protein, RC1 was discovered as the second crucial clue for the origin tracing of SARS-CoV-2. Although RC0, encoding the junction furin cleavage site, remains undetected outside of the SARS-CoV-2 genome, Feuang of Laos is the sole place where eight of the nine wild-type features (RC1-7 and *ORF8*) have been detected.

## Introduction

Since the emergence of SARS-CoV-2 [1], its genetic and geographical origins remain unclear, despite extensive research efforts. Several genomes of *Sarbecovirus* (also referred to as coronaviruses of betacoronavirus subgroups B [2]) have provided significant clues for tracing the geographical origin of SARS-CoV-2. Among them, the genome of a notable strain RaTG13 [3], isolated from intermediate horseshoe bats (*Rhinolophus affinis*), gained prominence shortly after the COVID-19 outbreak. Initially, RaTG13 was regarded as the closest relative of

**Competing interests:** The authors have declared that no competing interests exist.

SARS-CoV-2, primarily due to its high nucleotide (nt) identity with SARS-CoV-2 at the genome level, until the discovery of a more closely related strain, BANAL-20-52 [4]. BANAL-20-52 exhibits a higher nt identity of 96.85% (28940/29881) with SARS-CoV-2 compared to RaTG13, which has a nt identity of 96.13% (28720/29875). In the analysis of the BANAL-20-52 genome, researchers placed importance on the genomic features, particularly those encoding the receptor-binding domain (RBD) and the junction furin cleavage site (FCS) in the Spike (S) protein [5], rather than relying solely on simple analyses based on nt identities. However, the relashionship between BANAL-20-52 and SARS-CoV-2 was not determined, as researchers still grappled with the complexity of analyzing recombination regions. The discovery of BANAL-20-52 significantly contributed to expanding our understanding of the location where SARS-CoV-2 originated. Previously, this location was pinpointed to Tongguan town in Yunnan province of China where RaTG13 had been discovered, approximately 1,533 km from the outbreak site, as RaTG13 had been regarded as the closest relative. With the discovery of BANAL-20-52 in a region in Laos, approximately 518 km from Tongguan town, the understanding of the SARS-CoV-2's geographical origin was expanded to include Southeast Asia and South China [4], posing new challenges.

The genomes of these *Sarbecovirus* strains have been and will continue to be amassed to provide potential for unraveling the genetic origin of SARS-CoV-2. However, the data analysis is impeded by two intertwined obstacles: the high frequency of coronaviruse (CoV) recombination and the limitations of phylogenetic analysis [6]. Initially, many researchers focused on phylogenetic analysis to reconstruct the evolutionary history. However, they encountered the first obstacle—the high frequency of CoV recombination. Subsequently, they shifted their focus to the identification of potential donor strains by analysis of recombinant regions. This shift was based on the fundamental understanding that all CoV strains have complex evolutionary histories and are recombinant strains to which different progenitors (*i.g.*, donor strains) contribute. If researchers persist in tracing SARS-CoV-2's genetic origin by piecing together a puzzle, they will inevitably expand the list of potential donor strains as more *Sarbecovirus* genomes become available. However, without setting criteria, the identified potential donor strains may be isolated from distant locations, making it increasingly challenging to determine the geographical origin of SARS-CoV-2. Finally, they may consistently return to an old conclusion that there is no determined origin to SARS-CoV-2, and the emergence of SARS-CoV-2 is a simply evolutionary and selective process in which chance and environment play a key role [7]. Despite the numerous recombination events occurred in the evolutionary history, it is still possible to identify a few key or even crucial recombination events to outline the latest evolutionary history of SARS-CoV-2 before the COVID-19 outbreak, without necessarily knowing about other recombination events that occurred before. However, this requires substantial fundamental research to deepen our understanding of CoV recombination.

By analysis of genomes sequences and Nanopore RNA-seq data, we conducted a series of fundamental research [2, 8, 9] and proposed: (1) the replication, transcription and recombination of CoVs share the same molecular mechanism, which inevitably causes CoV outbreaks; (2) recombination, RBDs, junction FCSs, *ORF8*s [10] and other factors are key contributors to extraordinary transmission, virulence and host adaptability of betacoronavirus; (3) the strong recombination ability of CoVs combined with other key factors, generates multiple recombinant strains, two of which evolved into SARS-CoV and SARS-CoV-2, resulting in the SARS and COVID-19 pandemics; and (4) as the most important feature of SARS-CoV-2, the junction FCS is the crucial clue for future studies of its origin and evolution, much like the *ORF8* gene for SARS-CoV. Based on these new insights, we proposed that the genesis of SARS-CoV-2 was determined by several key or crucial genomic features, integrated through several key or crucial recombination events. By examing these genomic features, the close relatives of

SARS-CoV-2 can be identified. Through analyzing the genome data of these close relatives, the latest evolutionary history of SARS-CoV-2 can be outlined to completely understand the COVID-19 outbreak.

In the subsequent study [2], we identified nine key genomic features of *Sarbecovirus*, as large-scale mutations predominantly concentrate in these genomic regions. After further analysis of the mutations, we proposed that the nine key genomic features observed in wild types of SARS-CoV-2 remain detectable in one or several *Sarbecovirus* populations in the region where SARS-CoV-2 originated. This hypothesis allows for tracing SARS-CoV-2's genetic and geographical origins by only detection of the nine wild-type features before piecing together the whole puzzle. The first key genomic feature, designated as RC0, was reported in one of our previous studies [5], where wild-type RC0 was also identified to encode the junction FCS "RRA**R**" in SARS-CoV-2. Specifically, we asserted that SARS-CoV-2 is the only one among all reported *Sarbecovirus* strains to possess this distinctive feature. Despite efforts, wild-type RC0 remains undetected outside of the SARS-CoV-2 genome, as confirmed by all new data. Unfortunately, our research proposal was not funded. Then, we sought to validate our hypothesis using new genome data of *Sarbecovirus* strains reported after the COVID-19 outbreak. The primary results of the present study included the identification of three close relatives of SARS-CoV-2 (BANAL-20-52 [4], Rp22DB159 [11], and S18CXBatR24 [12]) and the successful detection of seven out of the nine wild-type features in the three close relatives from Laos, Vietnam, and Yunnan of China, respectively. Particularly, a key genomic feature, designated as RC1, in wild genotype, was later detected in a bat *Sarbecovirus* population BANAL-20-52 belonging to. Encoding a segment of the *NSP3* protein, RC1 was discovered as the second crucial clue for the origin tracing of SARS-CoV-2. Finally, integration of the new crucial evidence connected previously sparse and isolated evidence into a coherent chain. Based on this chain, the present study aimed to outline the latest evolutionary history of SARS-CoV-2 before the COVID-19 outbreak, leading to fully understand the orgin of SARS-CoV-2.

## Materials and methods

The genus *Betacoronavirus* consists of five main subgenera *Embecovirus*, *Sarbecovirus*, *Merbecovirus*, *Nobecovirus*, and *Hibecovirus* which were defined as subgroups A, B, C, D, and E, respectively. In the previous study [2], 1,265 genome sequences of *Betacoronavirus* that had been generated and submitted before the COVID-19 outbreak were downloaded from the NCBI Virus database (https://www.ncbi.nlm.nih.gov/labs/virus). Among these genomes, 292 belonging to *Sarbecovirus*, were used in the present study for analysis. The present study compiled a total of complete or partial genomic sequences from 156 bat *Sarbecovirus* strains from three independent studies [4, 11, 12] into a dataset (**S1 Text**). Among the genomes of 156 bat *Sarbecovirus* strains, nine were intensively analyzed, including SARS-CoV-2 (GenBank: MN908947), BANAL-20-52 (GenBank: MZ937000), Rp22DB159 (GenBank: OR233302), S18CXBatR24 (GenBank: OP963576), RaTG13 (GenBank: MN996532), BANAL-20-103 (GenBank: MZ937001), BANAL-20-236 (GenBank: MZ937003), BANAL-20-116 (GenBank: MZ937002), and BANAL-20-247 (GenBank: MZ937004). The genomes of RmYN01 and RmYN02 (GISAID: EPI_ISL_412976 and EPI_ISL_412977) were reassembled in our previous study [2] to improve their quality. MP789 (GenBank: MT121216) and PCoV_GX-P1E (GenBank: MT040334) are pangolin bat *Sarbecovirus* strains used as control.

The software VirusDetect [13] was used to detect and assemble the viruse contigs using RNA-seq data. The software Fastq_clean [14] was used for RNA-seq data cleaning and quality control. Sequence alignment was performed using the Bowtie v0.12.7 software with paired-end alignment allowing 3 mismatches. Multiple alignment was performed using MAFFT

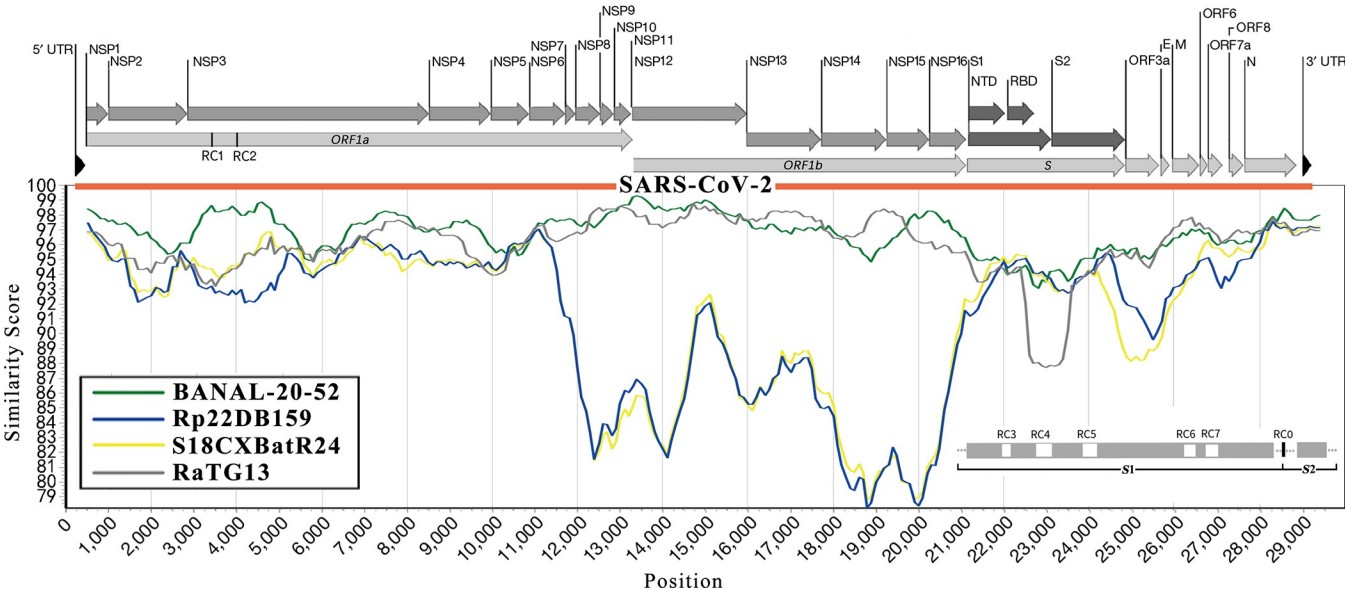

**Fig 1. Key genomic features in *Sarbecovirus*.** Based on our previous study [8], the SARS-CoV-2 genome consists of 10 genes, *ORF1a*, *ORF1b*, *S*, *ORF3a*, *E*, *M*, *ORF6*, *ORF7a*, *ORF8*, and *N(ORF9a)*. The NSP1-11 proteins are encoded by *ORF1a*, while the NSP12-16 proteins are encoded by *ORF1b*. The first key genomic feature, designated as recombinant region 0 (RC0) encodes a segment containing a furin cleavage site (FCS) "RRAR" in the junction region between S1 and S2 subunits (junction FCS) of the spike protein. RC1-8 encode peptides or segments that are located at NSP3:165–206 aa, NSP3:394–412 aa, S:67–78 aa, S:137–164 aa, S:239–262 aa, S:438–452 aa, S:468–486 aa, and S:1–34 aa. RC0-7 and the *ORF8* gene were identified as the key genomic features in our previous study [2]. The S protein is cleaved into two subunit S1 and S2 for receptor binding and membrane fusion. S1 has two domains, NTD (S:14–303 aa) and RBD (S:319–541 aa), which are annotated affiliated as with the reference sequence (Uniprot: P0DTC2). SARS-CoV-2 (red) is the Wuhan-2019 strain used as the reference. BANAL-20-52 (green), Rp22DB159 (blue), and S18CXBatR24 (yellow), isolated from malayan horseshoe bats (*Rhinolophus malayanus*) in Laos, least horseshoe bats (*Rhinolophus pusillu*) in Vietnam, and marshall's horseshoe bats (*Rhinolophus marshalli*) in Yunan of China, respectively (**Materials and Methods**). RaTG13 was isolated from intermediate horseshoe bats (*Rhinolophus affinis*) in Yunan of China.

v7.490. Statistics and plotting were conducted using the software R v2.15.3 with the Bioconductor packages [15]. The analysis in Fig 1 was performed using the program SimPlot [16] v3.5.1 with the Kimura two-parameter model, a window size of 1,001 base pairs, a step size of 100 base pairs, a transition/transversion rate (T/t) of 2.0, and a Gap/Strip parameter of "on". All other data processing were carried out using in-house Perl scripts. Close relatives of SARS-CoV-2 were identified according to the criteria: a toal of 343-aa residues of peptides or protein segments encoded by the nine key genomic features are required to be 100% covered with at least 90% identitiy. Homology models of the complexes between the RBDs of S18CXBatR24, Rp22DB159 and hACE2 were constructed using the X-ray structure of the complex between the SARS-CoV-2 RBD and human ACE2 (PDB: 6M0J).

## Results and discussion

### Nine key genomic features of *Sarbecovirus*

In our previous study [2], key genomic features were identified through the analysis of mutation sites in *Sarbecovirus*. Firstly, mutations were identified in the *Sarbecovirus* genomes, and then classified into two categories: small- and large-scale mutations. Small-scale mutations included single nucleotide substitutions (SNSs) and small insertions or deletions (InDels), whereas large-scale mutations included segment substitutions and large InDels involving at least 10 nucleotides. Secondly, by small InDels at six specific sites, *Sarbecovirus* was divided into two classes: the SARS1 class, which included SARS-CoV (from patients) and SARS-like CoV (from animals), and the SARS2 class, which included SARS-CoV-2 (from patients) and

SARS2-like CoV (from animals). Thirdly, large-scale mutations occurring within and between SARS1 and SARS2 classes were detected using 292 *Sarbecovirus* genomes [2]. Finally, we found that almost all the detected large-scale mutations concentrated in nine genomic regions, identified as nine key genomic features. In addition, the genotypes of genomic features observed in the SARS-CoV-2 genome (GenBank: MN908947) were designated as wild-type.

Here, nine key genomic features (designated RC0-7 and *ORF8*) are introduced (**Fig 1**). The first key genomic feature, designated as recombinant region 0 (RC0), was identified as the first crucial clue for the origin tracing of SARS-CoV-2 [2]. Wild-type RC0, encoding the junction FCS of SARS-CoV-2 [5], remains undetected outside of the SARS-CoV-2 genome. Another notable key genomic feature is the *ORF8* gene [10], which is the crucial clue for the origin tracing of SARS-CoV; however, it has no special significance for SARS-CoV-2. The other seven key genomic features were designated as RC1-7, which encode 12- to 42-amino-acid(aa) peptides or protein segments. RC1-2. RC3-5, and RC6-7 encode peptides in the *NSP3* protein [17], N-terminal domain (NTD) and RBD of the S1 subunit, respectively (**Fig 1**). In addition, a new genomic feature, designated as RC8, was identified during the present study. RC8 encodes the first approximately 30 aa residues, including the signal peptide of the S protein, and exhibites even higher diversity than RC1-7, as evidenced by the detection of many frame-shift mutations within it. Although the diversity of RC0-8 and *ORF8* was validated using new genome data (**Materials and Methods**), only nine (RC0-7 and *ORF8*) were used in our data analysis. Notably, a comprehensive study of 4,976,200 SARS-CoV-2 variants [18] revealed that InDels are still highly frequently occurring in RC3-5 during the COVID-19 pandemic, while RC1-2 and RC6-7 remain relatively conservative, with no highly frequent InDels detected in them.

Our previous study [2] revealed that the peptides or protein segments encoded by RC0-7 are disordered in their secondary structures, and their specific functions remain largely unknown, except for RC6 and RC7. By analyzing the structure data [2], we predicted that the peptides encoded by RC6 and RC7 contain all key aa residues in the interaction of RBD with its receptor angiotensin converting enzyme 2 (ACE2). As almost all the detected large-scale mutations in RBD concentrated in RC6 and RC7, we designated them as key genomic features, rather than the entire RBD. These disordered peptides or protein segments, also referred to as hypervariable regions (HVRs), are typically involved in protein-protein interactions (PPIs). Thus, they may have regulatory rather than essential roles in the CoV life cycle, allowing them to tolerate numerous large-scale mutations. In our previous study [2], we linked the mutations in HVRs, especially the nine key genomic features, to the adaptability of *Sarbecovirus* for new hosts or even the expansion of their host range [19]. However, it is unclear which specific mutations in RBDs enhance their interactions with host receptors and which specific mutations contribute to immune evasion, as the mechanisms underlying the adaptability remain unknown. In the comprehensive study of 4,976,200 SARS-CoV-2 variants [18], reseachers also identified HVRs encoded by RC1-5, along with additional HVRs. In addition, they proposed that the increased frequency of indels, the non-random distribution of them and their independent co-occurrence in several variants of concern (VOCs) is another mechanism of response to elevated global population immunity [18]. Unfortunately, they did not reveal the underlying mechanisms at the molecular level, probably because they did not investigate the connection between mutations and the jumping transcription of CoVs [8, 9].

Integrating the results from our previous study [2] and the previous study [14], several important findings were obtained, including: (1) CoV HVRs experience highly frequent InDels, compared to other genomic regions; (2) deletions are more prevalent than insertions within HVRs; (3) several notable large deletions (longer than 10 nucleotides) in HVRs of SARS-CoV and SARS-CoV-2 were reported in attenuated strains [20–22]; and (4) each of the

nine key genomic features has a few distinct genotypes. According to our model explaining the jumping transcription of CoVs [8, 9], the typical occurrence of jumping transcription relies on two types of transcription regulatory sequences (TRSs) successfully matching: leader TRSs (TRS-Ls) and body TRSs (TRS-Bs). Recombination occurs when the antisense sequences of TRS-Bs anneal or connect to regions containing TRS-like sequences, rather than TRS-Ls in the RNA synthesis process. Although CoV recombination is highly frequent, most of recombination events result in small-scale mutations, defined as the type 1 recombination, while only a few of them result in large-scale mutations, defined as the type 2 recombination. One genotype of a key genomic feature changes into another genotype relying on type 2 recombination, rather than evolves into another genotype through gradual accumulation of type 1 recombination. A large InDel results from a single recombination event, rather than gradual accumulation of small-scale mutations. As a result, the genotypes of the nine key genomic features observed in wild types of SARS-CoV-2 remain unchanged or undergo only minor modifications after a short period of evolution. Consequently, all the nine wild-type features remain detectable in one or several *Sarbecovirus* populations after the COVID-19 outbreak, particularly in the region where SARS-CoV-2 originated. This hypothesis allows for tracing SARS-CoV-2's genetic and geographical origins by only detection of the nine wild-type features before piecing together the whole puzzle.

### Three close relatives of SARS-CoV-2

The relationship between newly discovered *Sarbecovirus* strains and SARS-CoV-2 were simply measured by their nt identities at the genome level. Regarding genomic features, most of previous studies focused on the RBD corresponding to RC6-7, while only a few of them examined the junction FCS corresponding to RC0 [5]. RaTG13 was identified as the closest relative to SARS-CoV-2, soon after the COVID-19 outbreak; however, its genomic features were insufficient to fully elucidate the origin of SARS-CoV-2. Two years later, a closer relative BANAL-20-52 was discovered with a higher nt identity with SARS-CoV-2. Furthermore, BANAL-20-52 shares almost identical RBDs and *ORF8* with SARS-CoV-2, yet it lacks the junction FCS as all other bat *Sarbecovirus* strains. According to our proposal, the researchers who discovered BANAL-20-52 examined only three of the nine genomic features (RC6-7 and *ORF8*). By examining the nine key genomic features (**Fig 2**), it was found that BANAL-20-52 harbor seven wild-type features (RC2-7 and *ORF8*). Subsequently, we expanded our examination to include new genome data of *Sarbecovirus* submitted after the COVID-19 outbreak, by setting the criteria (**Materials and Methods**). As a result, a total of three strains (**Fig 1**) were identified as close relatives of SARS-CoV-2, harboring seven wild-type features (**Table 1**). They are BANAL-20-52 [4], Rp22DB159 [11], and S18CXBatR24 [12], isolated from malayan horseshoe bats (*Rhinolophus malayanus*) in Laos, least horseshoe bats (*Rhinolophus pusillu*) in Vietnam, and marshall's horseshoe bats (*Rhinolophus marshalli*) in Yunan of China, respectively.

By examining the nine key genomic features, many potential "donor strains" were excluded from close relatives, although a certain of them share high nt identites (>90%) with SARS-CoV-2 at the genome level as the three close relatives. Despite harboring five wild-type features (RC2-5 and ORF8), RaTG13 was still excluded (**Table 1**), according to our criteria (**Materials and Methods**). Before the present study, there was no consensus understanding of the relationship between RaTG13 and SARS-CoV-2, mainly because of significant differences observed in their RBDs, equivalent to RC6 and RC7 (**Fig 2B**). These differences led to a speculation that the emergence of SARS-CoV-2 may have depended on recombination or natural selection to enhance its RBD affinity to human ACE2 [4]. Later, the speculation was challenged by the discovery of BANAL-20-52 and was finally refuted by our identification of the three

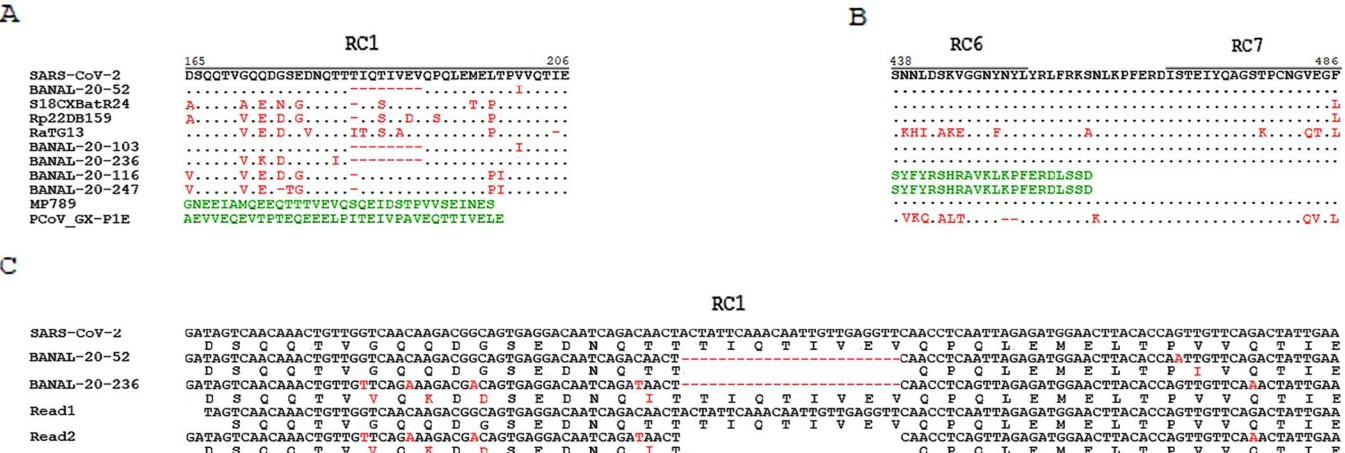

**Fig 2. Crucial evidence for origin tracing of SARS-CoV-2.** SARS-CoV-2 is the Wuhan-2019 strain used as the reference. BANAL-20-52, S18CXBatR24, Rp22DB159, RaTG13, BANAL-20-103, BANAL-20-236, BANAL-20-116, and BANAL-20-247 are bat *Sarbecovirus* strains, while MP789 and PCoV_GX-P1E are pangolin bat *Sarbecovirus* strains. The accession numbers of their genomes were provided in the **Materials and Methods.** As three of the nine key genomic features in *Sarbecovirus* (**Fig 1**), RC1 and RC6-7 encode peptides in the NSP3 protein and RBD of the S1 subunit in the S protein, respectively. The dots indicate aa or nt residues same to the reference. The mutant aa or nt residues are indicated by red color, while the mutant segments (different genotypes) are indicated by green color. Read1 (ID: 14445348.2) and Read2 (ID: 14755068.2) are two reads from the NGS data (SRA: SRR22719882) of the sample containing BANAL-20-236. Read1 and Read2 represent reads which were aligned to wild-type RC1 in SARS-CoV-2 and those which were aligned to mutant RC1 in BANAL-20-236, respectively, as they entirely covered wild-type RC1 and mutant RC1, respectively.

close relatives which harbor RBDs with similar affinity to human ACE2 as SARS-CoV-2. Several pangolin *Sarbecovirus* strains had been considered as "donor strains", also because they harbor RBDs with similar affinity to human ACE2 as SARS-CoV-2. Among them, MP789 shares almost identical RC6 and RC7 with SARS-CoV-2 (**Fig 2B**). However, MP789 and another strain PCoV_GX-P1E [23], harboring only two (RC6 and RC7) and one (*i.e.*, RC7) wild-type features respectively, were excluded from close relatives. Two notable strains

**Table 1. Key genomic features examined in *Sarbecovirus* strains.**

| | NSP3 | | S | | | | | ORF8 |
|---|---|---|---|---|---|---|---|---|
| | RC0 | RC1 | RC2 | RC3 | RC4 | RC5 | RC6&7 | ORF8 |
| SARS-CoV-2 | Wild | Wild | Wild | Wild | Wild | Wild | Wild | Wild |
| BANAL-20-52 | InDel | InDel+1 | 2 | 1 | 1 | 0 | 0 | 5 |
| Rp22DB159 | InDel | 10 | 3 | 2 | 0 | 0 | 1 | 6 |
| S18CXBatR24 | InDel | 9 | 3 | 2 | 0 | 0 | 1 | 4 |
| RaTG13 | InDel | 10 | 5 | 1 | 0 | 0 | 15 | 6 |
| BANAL-20-103 | InDel+1 | InDel+1 | 1 | Mut | 12 | 13 | 0 | 5 |
| BANAL-20-236 | InDel | InDel+4 | 4 | Mut | 12 | 13 | 0 | 3 |
| BANAL-20-116 | InDel+9 | 8 | 4 | Mut | Mut | Mut | Mut | Mut |
| BANAL-20-247 | InDel+9 | 9 | 4 | Mut | Mut | Mut | Mut | Mut |
| MP789 | InDel+3 | Mut | Mut | Mut | 11 | 13 | 0 | InDel+6 |
| PCoV_GX-P1E | InDel+10 | Mut | Mut | Mut | 5 | 4 | 14 | * |

SARS-CoV-2 is the Wuhan-2019 strain used as the reference. BANAL-20-52, S18CXBatR24, Rp22DB159, RaTG13, BANAL-20-103, BANAL-20-236, BANAL-20-116, and BANAL-20-247 are bat Sarbecovirus strains, while MP789 and PCoV_GX-P1E are pangolin bat Sarbecovirus strains. The accession numbers of their genomes were provided in the Materials and Methods. RC0-7 and the *ORF8* gene were identified as the key genomic features in our previous study [2]. Wild and Mut (green color) represents wild-type and mutant genotypes. InDel represents insertions or deletions, while the number behind InDel indicates how many aa residues have been substituted. * stop codons are present in the ORFs.

RmYN01 and RmYN02, harboring only 0 and two (RC1 and RC2) wild-type features respectively, were also excluded, although nature insertions in their RC0s had been reported in a previous study [24]. Furthermore, RmYN02 possesses a unique recombinant *ORF8* identified in our previous study [2], which may have resulted from a rare recombination event between *Sarbecovirus* strains of the SAR1 class and those of the SAR2 class. In a broader sense, almost all *Sarbecovirus* strains can be considered as potential "donor strains" contributing to the genesis of SARS-CoV-2, given their high nt identities with SARS-CoV-2 at the genome level or even in a very small genomic region. Identification of all the recombination events involving these "donor strains" in the evolutionary history could not provide significant clues for the origin tracing of SARS-CoV-2. Instead, comparing the genomes of the three or more close relatives may provide insights into the latest recombination events, which made key or even crucial contributions to the genesis of SARS-CoV-2.

While Rp22DB159 and S18CXBatR24 exhibit slightly lower nt identities (92.30% and 92.19%,) with SARS-CoV-2, respectively, compared to RaTG13 (96.13%), they possess two additional key genomic features (RC6 and RC7) in wild genotypes (**Fig 2B**), absent in RaTG13. The three close relatives (BANAL-20-52, Rp22DB159, and S18CXBatR24) habor wild-type NTDs and RBDs. Particularly, the genomic regions encoding the RBDs of the three close relatives have very high nt identities of 93.87% (628/669), 93.87% (628/669) and 93.57% (626/669) with SARS-CoV-2, respectively, which were wild-types. Compared to SARS-CoV-2, there are nine single aa substitution sites in RBDs of the three close relatives (**S1 Text**), including four common aa substitutions (A372T, Q498H, H519N, and V534I) shared by the three relatives, two aa substitutions (E324Q and F486L) specific to Rp22DB159 and S18CXBatR24, two aa substitutions (R346T and I402V) specific to BANAL-20-52, and a unique aa substitution N501Y present only in S18CXBatR24. Among these single aa substitutions, only F486L, Q498H, and N501Y were detected to be associated with the 17 aa residues of the RBD that interact with human ACE2 [4]. By comparing the RBD-ACE2 complex structures of SARS-CoV-2 with those of the three close relatives (**Materials and Methods**), it can be observed that F486L only causes a slight decrease in the hydrophobic surface at position 486, which may not significantly impact the interaction between the RBD and human ACE2. The notable Q498H, shared by three close relatives, was reported to enhance the affinity of the SARS-CoV-2 RBD to human ACE2, and also to be involved in the host range expansion of SARS-CoV-2 and SARS2-like CoVs [4]. N501Y, along with N501T are two common mutations at the N501 position present in at least five VOCs, including Alpha (B.1.1.7), Beta (B.1.351), Gamma (P.1), Delta (B.1.617.2) and Omicron (B.1.1.529). Detected in approximately 45.7% and 0.00043% of SARS-CoV-2 variants, N501Y and N501T may enhance and reduce the affinity of the SARS-CoV-2 RBD to human ACE2, respectively [25]. According to a comprehensive analysis with validation by experiments [4], BANAL-20-52 possesses a RBD with enhanced affinity to human ACE2, primarily due to Q498H. Therefore, Rp22DB159, which harbors Q498H, possesses a RBD with the similar affinity as BANAL-20-52, while S18CXBatR24 could possess a RBD with even higher affinity, as it harbors both Q498H and N501Y.

The seven wild-type features in the three close relatives preliminarily validated our hypothesis that all key genomic features of SARS-CoV-2 in wild genotypes remain detectable in one or several *Sarbecovirus* populations in the region where SARS-CoV-2 originated. Unexpectedly, the three close relatives haboring seven wild-type features were identified in samples independently collected from locations distant from each other. Hence, future research should have focused on the search for the two remaining key genomic features (RC0 and RC1) in wild genotypes within the three regions where the three close relatives had been discovered. However, the crucial clue –wild-type RC0 remains undetected outside of the SARS-CoV-2 genome, which is consistent with our previous prediction [5] that it may have been acquired in

intermediate hosts or humans. Fortunately, RC1 was identified as the second crucial clue during the present study, as RC1 and its encoded segment exhibited large differences between SARS-CoV-2 and its three close relatives (**Fig 2A**), compared to the seven key genomic features (RC2-7 and *ORF8*) that are almost identical between SARS-CoV-2 and its three close relatives. Wild-type RC1 encodes a 42-aa segment (NSP3: 165–206 aa) localized in an HVR of NSP3. This HVR was annotated as spanning 113–206 aa residues of the NSP3 protein in SARS-CoV-2, by analysis of the structures of its neighboring domains (**Materials and Methods**). The similarities between the 42-aa segment of SARS-CoV-2 and their homologs in the three close relatives are even lower than that of RmYN02, which harbors seven single aa substitutions. The homologs in Rp22DB159 and S18CXBatR24 harbor nine and ten single aa substitutions, respectively, while that in BANAL-20-52 exhibits one single aa substitution and an 8-aa deletion—TIQTIVEV (**Fig 2A**). Using the 42-aa segment to search for closer homologs, numerous homologs within the realm of animal *Sarbecovirus* were obtained from NCBI NT and NR databases. Among them, the closest homolog containing six single aa substitutions was identified in Rp22DB167, which had been isolated from the collection site of Rp22DB159. However, the quality of the Rp22DB167 genome is considerate low. All other homologs exhibit more than six single aa substitutions compared to the wild-type 42-aa segment.

## The discovery of wild-type RC1 as crucial evidence

Following the identification of RC0, RC1 was identified as the second crucial clue during the present study, its uniqueness underscored its particular significance. Notably, the 8-aa deletion in *NSP3* of BANAL-20-52 (corresponding to a 24-nt deletion in RC1) is analogous to the 4-aa deletion (corresponding to a 12-nt deletion in RC0) resulting in loss of the junction FCS in all other SARS-2 like strains [5]. The RC1 of BANAL-20-52 contains only one single nucleotide polymorphism (SNP) and a 24-nt deletion, while those in Rp22DB159 and S18CXBatR24 contain nine and 11 SNPs, respectively (**Fig 2A**). According to our findings (**Described above**), the 24-nt deletion in RC1 or the 12-nt deletion in RC0 could have resulted from a single recombination event, rather than gradual accumulation of small-scale mutations. This suggested that the RC1 of BANAL-20-52 had diverged from that of SARS-CoV-2 relatively late, making the detection of wild-type RC1 highly probable in a *Sarbecovirus* population BANAL-20-52 belonging to. Further research should have focused on the search for wild-type RC1 within the region where BANAL-20-52 had been discovered, using additional genome data, especially of bat individuals or populations. Given the unavailability of additional genomes, we endeavored to search for it using the Next-Generation Sequencing (NGS) data of *Sarbecovirus* strains collected in the previous study [4], where BANAL-20-52 was reported. We believed that wild-type RC1 could be detected as low-depth contigs from minor genotypes, which might have been misassembled due to the limitations of the short reads produced by NGS technologies. In a sense, each viral genome sequence assembled using the NGS data is a mosiac genome sequence. It is assembled as a consensus representation of the genetic diversity within a virus population, reflecting the predominant genotypes at each position. However, low-depth contigs from minor genotypes, absent in the genome, are usually neglected for analysis. Although all genome sequences of individuals in a viral population can be obtained using advanced sequencing technologies such as PacBio and Nanopore DNA-seq [26], the resulting data typically includes sequences from major genotypes, with the risk of missing the sequences from minor genotypes.

As expected, wild-type RC1 was finally detected, albeit in the sample containing BANAL-20-236, rather than that containing BANAL-20-52. In the NGS data (SRA: SRR22719882) of the sample containing BANAL-20-236, numerous reads longer than 100 bp were aligned to

the mutant RC1 in BANAL-20-236, while a few reads were aligned to wild-type RC1 in SARS-CoV-2 (**Fig 2C**). Among these reads, 56,168 reads entirely covered mutant RC1, while only two reads entirely covered wild-type RC1. The uniqueness of wild-type RC1 ruled out the possibility that its reads resulted from other sources. The reads aligned to wild-type RC1 had not been assembled into the BANAL-20-236 genome, due to their lower abundance compared to those aligned to mutant RC1. At first glance, this suggested that a *Sarbecovirus* population, BANAL-20-236 belonging to, possessed eight wild-type features (RC1-7 and *ORF8*). However, based on similarities measured by the seven key genomic features, all five *Sarbecovirus* strains (**Table 1**) reported in the previous study [4] can be classified into three populations and they are: (1) the BANAL-20-236 population including at least the BANAL-20-103 and BANAL-20-236 strains; (2) the BANAL-20-116 population including at least the BANAL-20-116 and BANAL-20-247 strains, and (3) the BANAL-20-52 population including at least itself. Among these strains, BANAL-20-52 is the closest relative of SARS-CoV-2, while BANAL-20-236 exhibits the largest differences from SARS-CoV-2 in their genomes. It is more likely that wild-type RC1 was acquired by BANAL-20-52 than BANAL-20-236 for the genesis of SARS-CoV-2. Therefore, wild-type RC1 may also be present in the BANAL-20-52 populaiton as a minor genotype. If so, a potential closest relative could be assembled using the NGS data of BANAL-20-52 or other samples. This closest relative, named BANAL-RC1, harboring eight wild-type features (RC1-7 and *ORF8*), is even closer to SARS-CoV-2 than BANAL-20-52 which harbors seven wild-type features. It is conceivable that BANAL-RC1 was previously and is still present within this *Sarbecovirus* population. However, the NGS data of BANAL-20-52 was not available for validation. Nevertheless, wild-type RC1 was detected in a large bat *Sarbecovirus* population BANAL-20-52 and BANAL-20-236 belonging to, which dispersed over a region with a diameter at least 2.22 km, harboring eight wild-type key genomic features. The collection sites of BANAL-20-52 and BANAL-20-236 were localized in caves named Pha Nouk Kok (18.53N, 101.98E) and Pha Tong (18.55N, 101.98E) with a short distance of 2.22 km. In addition, both of BANAL-20-52 and BANAL-20-236 had been detected in the samples, isolated from malayan horseshoe bats (*Rhinolophus malayanus*).

## The origin of SARS-CoV-2 in bats

The data of the three close relatives were sourced from three independent studies [4, 11, 12], respectively. These studies contributed a total of complete or partial genomic sequences from 156 bat *Sarbecovirus* strains, which were compiled into a dataset containing a wealth of geographic information for our further analysis (**Materials and Methods**). Among the 156 bat *Sarbecovirus* strains, 111 were used in the previous study [11]. Based on a phylogeographic analysis, the previous study [11] proposed a divergent evolution between sarbecoviruses from subtropical northern Vietnam and those from tropical southern Vietnam, emergence of SARS-CoV in horseshoe bats from northern Yunnan and emergence of SARS-CoV-2 in horseshoe bats from southern Yunnan. Although the phylogeographic analysis applied a new method, named coloured genomic bootstrap (CGB) barcode, it still exhibited two limitations. The first pertains to the absence of S18CXBatR24 in the analyzed dataset. This omission renders misleading information that none bat *Sarbecovirus* strains from Yunnan of China harbor wild-type RC6 and RC7. The second limitation lies in the inherent nature of phylogeographic analysis, which can only provide estimates of the relative relationships between different strains, requiring concrete evidence, like wild-type RC1, to determine the directionality of these relationships. Indeed, using phylogeographic analysis for the origin tracing of SARS-CoV-2 does not yield additional information beyond what is obtained from the simple analyses based on nucleotide identities.

Different from the previous study [11], we used the geographical information associated with the 156 bat *Sarbecovirus* strains to delineate the distribution of the seven wild-type key genomic features across Southeast Asia and South China (**Fig 3**). Generally speaking, most of the bat *Sarbecovirus* strains did not harbor more than three wild-type key genomic features. Notably, only the three close relatives and RaTG13 harbored seven and five wild-type key genomic features, respectively. These new findings were consistent with our previous conclusion that the genesis of SARS-CoV-2 was determined by several key or crucial genomic features, integrated through several key or crucial recombination events. Wild-type RC6 and RC7 are not rare among bat *Sarbecovirus*, suggesting that they might not have made crucial contributions to the genesis of SARS-CoV-2. However, possessing wild-type RBDs, equivalent to harboring wild-type RC6 and RC7, is a basic requirement for bat *Sarbecovirus* strains to evolve into SARS-CoV-2. In contrast, wild-type RC0 and RC1 are rare, suggesting they may have made crucial contributions to the genesis of SARS-CoV-2. As wild-type RC1 was only detected in a bat *Sarbecovirus* population in Laos and wild-type RC0 remained undetected in bats, wild-type RC0 and RC1 may have been integrated into SARS-CoV-2 through the last and second-to-last type 2 recombination events, respectively. In addition, wild-type RC0 in SARS-CoV-2 may have been acquired in intermediate hosts or humans [5].

Despite wild-type RC0 not being detected outside of the SARS-CoV-2 genome, integration of the new crucial evidence (wild-type RC1) connected previously sparse and isolated evidence into a coherent chain, concentrating on the key genomic features and the three close relatives from Feuang, Vientiane province of Laos (Feuang), Dien Bien municipality of Vietnam (DB-3), and Chuxiong, Yunnan province of China (CX). It's highly improbable that these three close relatives independently evolved to harbor seven wild-type features as SARS-CoV-2. Instead, the origins of the three close relatives involved *Sarbecovirus* populations from Feuang, DB-3, and CX (**Fig 3**), suggesting a complex pattern of viral transmission and recombination events across various locations. Since wild-type RC1 was only detected in the *Sarbecovirus* population in Feuang, a putative direction of the SARS-CoV-2 genesis could be inferred from either DB-3 or CX to Feuang, representing Vietnam or Yunnan of China, to Laos (**Fig 3**). The genesis of SARS-CoV-2 likely involved a few key recombination events in limited *Sarbecovirus* groups within a limited number of regions. These key recombination events could have occurred in three bat hosts: malayan horseshoe bats (*Rhinolophus malayanus*), least horseshoe bats (*Rhinolophus pusillu*), and marshall's horseshoe bats (*Rhinolophus marshalli*). The last or second-to-last recombination events crucial to the genesis of SARS-CoV-2 could have occurred in malayan horseshoe bats in tropical regions, particularly Feuang or its vicinity.

The direction of the SARS-CoV-2 genesis between DB-3 and CX can not be determined by key genomic features, given that both Rp22DB159 from DB-3 and S18CXBatR24 from CX harbor seven wild-type features. Furthermore, Rp22DB159 and S18CXBatR24 share a nucleotide identity of 97.38%, which is the highest among all identities between the 156 strains. According to a previous study, *Sarbecovirus* genomes from lower latitudes have a stronger synonymous nucleotide composition (SNC) bias in favour of U nucleotides than those from higher latitudes. While, our study revealed an increase in A-content in *Sarbecovirus* genomes of S18CXBatR24, Rp22DB159, BANAL-20-52 from 29.7, 29.78 to 30.32%, suggesting the direction of the genesis of SARS-CoV-2 from CX to DB-3. Thus, we proposed that SARS-CoV-2 originated along a conceptual path from Yunnan, China to Vietnam, and then to Laos (**Fig 3**), acquiring crucial genomic features through crucial recombination events. However, it remains uncertain whether the driving force behind this path is attributable to climate or latitude, given that the three regions (CX, DB-3, and Feuang) span from subtropical to tropical climates, with latitudes ranging from 2000 m, 900 m to 3~400 m, respectively. Furthermore, the underlying mechanisams, particularly at the molecular level remain unknown. Notably, the increase

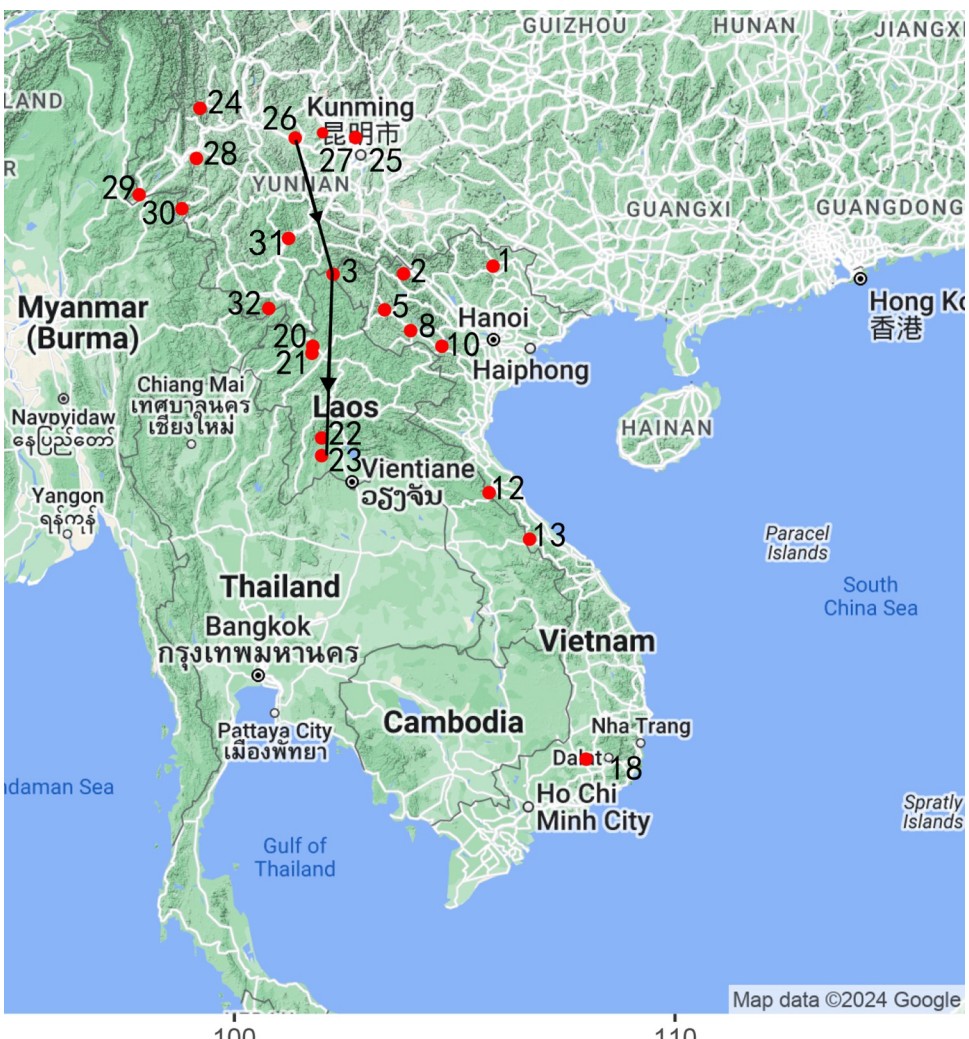

**Fig 3. The geographical distribution of studied *Sarbecovirus* strains.** Sample collection site 1–19 and 20–23 were reported in the previous study [4, 11], respectively. Site 4, 6, 7, 9, 11, 14, 15, 16, 17 and 19 were not shown in this figure, as no *Sarbecovirus* were detected in these locations. Three close relatives (BANAL-20-52, Rp22DB159, and S18CXBatR24) were discovered in Feuang, Vientiane province of Laos (Feuang), Dien Bien municipality of Vietnam (DB-3), and Chuxiong, Yunnan province of China (CX), respectively, while RaTG13 were discovered in Tongguan town, Yunnan province of China (TG). The distances between CX (25.22 N,101.38 E) and the outbreak site (30.61 N, 114.24 E) is 1,397 km (Not shown); the distance between TG (23.14 N, 101.23 E) and CX is 232 km; the distance between Feuang (18.53 N, 101.98 E) and CX is 746 km; the distance between DB-3 (22.39 N, 102.24 E) and CX is 327 km. The distances of Feuang-TG, Feuang-DB-3, DB-3-TG are 518, 430 and 133 km, respectively. Wild-type RC1 was only detected at site 20. We proposed that SARS-CoV-2 originated along a conceptual path from Yunnan, China to Vietnam, and then to Laos, acquiring crucial genomic features along the way. 3, 23, 26 and 31 are DB-3, Feuang, CX and TG. The latitudes and longitudes of 32 sample collection sites are provided in **S1 Text**.

in A-content has the potential to augment the NSP15 cleavage sites within the genome, possibly leading to an increased frequency of recombination events. According to our previous study [8, 9], the NSP15 cleavage sites contain U nucleotides at 5' ends, corresponding to A nucleotides in the TRS-like sequences and the frequency of recombination events dependents on the cleavage of synthesized RNAs at the NSP15 cleavage sites. Additionally, the similar A-content observed in SARS-CoV-2 and BANAL-20-52 (30.31% vs. 30.32%) could indicate shared evolutionary pressures or a common genetic background. In contrast, RaTG13 has a

lower A-content 29.94%, although it has the similar nt identities with SARS-CoV-2 as BANAL-20-52 (96.13% vs. 96.85%).

## Conclusions

In the present study, we conducted an integrative analysis of new genome data from bat *Sarbecovirus* strains reported after the COVID-19 outbreak. The primary results preliminarily validated our hypothesis that all key genomic features of SARS-CoV-2 in wild genotypes remain detectable in one or several *Sarbecovirus* populations. Particularly, wild-type RC0 remains detectable in *Sarbecovirus* populations beyond just bat hosts. Furthermore, these wild-type key genomic features may be present as minor genotypes. The most significant contribution lies in the detection of RC1 in wild genotype in a bat *Sarbecovirus* population BANAL-20-52 belonging to. Integration of the new crucial evidence has connected previously sparse and isolated evidence into a coherent chain. Based on this chain of evidence, a crucial short-term evolutionary history of SARS-CoV-2 before the COVID-19 outbreak was outlined, greatly advancing the origin tracing of SARS-CoV-2. This short-term history has been traced back to a bat *Sarbecovirus* strain BANAL-RC1 harboring eight wild-type key genomic features (RC1-7 and *ORF8*). Future research should prioritize detecting RC0 and RC1 to confirm the above results using additional samples collected from Laos, Vietnam, and Yunnan of China. As wild-type RC0 and RC1 may be present as minor genotypes, PCR amplification should be performed to enhance detection sensitivity, with the downstream gel electrophoresis, Sanger sequencing, or the NGS sequencing. By following this approach, we believe that the crucial short-term evolutionary history of SARS-CoV-2 will be completed, ultimately bridging the gap from BANAL-RC1 to SARS-CoV-2. Applying our methodologies, it would be intriguing to investigate whether the *ORF8* gene could be the crucial genomic feature finally integrated into SARS-CoV through recombination, leading to understand the SARS outbreak in 2003. In addition, the structural features of the protein regions encoded by RC0 and RC1 could be used in drug design to inhibit or even block the infection of SARS-CoV-2. For instance, angiotensin receptor blockers (ARBs) have been designed [27–31].

Unraveling the origin of SARS-CoV-2 is not only crucial for scientific knowledge, but also for improving global health security and pandemic preparedness. Unfortunately, the genetic and geographical origins of SARS-CoV-2 remain unknown, resulting in suspicions about its natural origin. Then, two main hypotheses have been addressed: 1) SARS-CoV-2 naturally spilled over from an animal reservoir to humans, possibly at the Huanan seafood market in Wuhan, China where the first cases were detected [32]; and 2) SARS-CoV-2 originated from a laboratory incident, potentially linked to coronavirus research at the Wuhan Institute of Virology [33]. Our findings disprove the second hypothesis, as none *Sarbecov*irus strains in Wuhan Institute of Virology or from Tongguan town in Yunnan province of China, particularly RaTG13, habor or acquired wild-type RC1, RC6, and RC7 together through recombination. However, due to the undetected wild-type RC0, which might have been acquired in intermediate hosts or humans [5], contributing to the genesis of SARS-CoV-2, our findings are insufficient to provide a conclusive answer to the first hypothesis. As Feuang of Laos is the sole place where eight of the nine wild-type features (RC1-7 and *ORF8*) have been detected, it is considered more probable than Wuhan to be the initial site where the ancestor of SARS-CoV-2 was transmitted from bats to intermediate hosts or directly to humans. This provides a clue to investigate where the animals infected by SARS-CoV-2 at the Huanan seafood market in Wuhan came from, addressing a question that the first hypothesis needs to answer. As an additional finding, several *Sarbecovirus* populations still harbor seven or eight other key genomic features in wild types, posing a certain risk of causing future pandemics. As there is no hope

for us to be funded to continue the relevant research, we hope that other scientists will find value in our findings and consider incorporating our insights into their own work.

## Supporting information

**S1 Text. All the genome IDs and the latitudes and longitudes of 32 sample collection sites.** (ZIP)

## Acknowledgments

We are grateful for the help from the following faculty members of College of Life Sciences at Nankai University: Wenjun Bu, Tao Zhang, Dawei Huang, Huaijun Xue, Qiang Zhao, Yanqiang Liu, Bingjun He, Wei Liu, and Zhen Ye. Special thanks should be given to Jinsong Shi from and National Clinical Research Center of Kidney Disease, Jinling Hospital and Alexandre Hassanin from Sorbonne Université. This manuscript has been submitted as a preprint on March 31st, 2024 at https://www.researchgate.net/ with acquisition of DOI: 10.13140/RG.2.2.23325.12008.

## Author Contributions

**Conceptualization:** Shan Gao.

**Data curation:** Yutong Guo, Haohao Yan, Liang Chen.

**Formal analysis:** Shunmei Chen, Shan Gao.

**Project administration:** Shan Gao.

**Software:** Cihan Ruan.

**Supervision:** Xin Li, Shan Gao.

**Validation:** Yongzhong Duan, Guangyou Duan, Jinlong Bei, Shan Gao.

**Visualization:** Jia Chang.

**Writing – original draft:** Shan Gao.

**Writing – review & editing:** Xin Li, Shan Gao.

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
