## [Decision Letter · Decision Letter 0]

12 Jul 2024

PONE-D-24-20443Emergence of crucial evidence catalyzing the origin tracing of SARS-CoV-2PLOS ONE

Dear Dr. Gao,

Thank you for submitting your manuscript to PLOS ONE. After careful consideration, we feel that it has merit but does not fully meet PLOS ONE’s publication criteria as it currently stands. Therefore, we invite you to submit a revised version of the manuscript that addresses the points raised during the review process.

We look forward to receiving your revised manuscript.

Kind regards,

AbdulAzeez Adeyemi Anjorin, Ph.D.

Academic Editor

PLOS ONE

3. We notice that your supplementary figures are uploaded with the file type 'Figure'. Please amend the file type to 'Supporting Information'. Please ensure that each Supporting Information file has a legend listed in the manuscript after the references list.

Reviewers' comments:

Reviewer's Responses to Questions

**Comments to the Author**

1. Is the manuscript technically sound, and do the data support the conclusions?

Reviewer #1: Yes

Reviewer #2: Yes

Reviewer #3: Partly

2. Has the statistical analysis been performed appropriately and rigorously? 

Reviewer #1: I Don't Know

Reviewer #2: Yes

Reviewer #3: N/A

3. Have the authors made all data underlying the findings in their manuscript fully available?

Reviewer #1: Yes

Reviewer #2: Yes

Reviewer #3: No

4. Is the manuscript presented in an intelligible fashion and written in standard English?

Reviewer #1: Yes

Reviewer #2: Yes

Reviewer #3: No

5. Review Comments to the Author

Reviewer #1: Chen S et al., analyzed the phylogenetic relationships among the Sarbecoviruses (in genus Betacoloronavirus) detected before outbreak of SARS-CoV-2 in the end of 2019 to trace the origin of SARS-CoV-2 in natural reservoir. The authors used 292 nucleotide sequences of Sarbecoviruses in Genbank including 156 bat-origine Sarbecoviruses for analysis.

There have been numerous epidemiological studies to find the origin of SARS-CoV2, but no clear conclusion has been reached. This study is unique in that it focuses on the nine genetic markers ‘RC0 - RC7 and ORF8’ rather than sequence similarity of whole gene taking into the consideration of frequence recombination of coronavirus genome.

Although the study did not reach out the direct origin of coronavirus strain connected to SARS-CoV-2, the analysis proposed that RC1, one of the genetic markers would contribute to trace the history of Sarbecoviruses in wild life.

Minor comment

Throughout the manuscript, descriptions seem subjective. It is recommended to describe result and discussion separately to differentiate the objective facts clearly.

Table 1. What are the numbers in the table?

Abstract. Please delete or modify the last sentence.

Is it reasonable to treat gene deletions and insertions equally?

Line 135. Please explain the RC0 ‘e.g. addition R insertion at position XX-XX’ here again.

Fig. 1. The position and scale of S1-S2 gene showing RC3-RC7 and RC0 (shown in right-bottom) was difficult to understand. Please make clear the relationship to the image of SARS-CoV-2 gene structure.

Line 136-138. Please explain the example of genetic characteristics of ORF8.

Lines 141-145. Why was RC8 was not included in the analysis? If not used for analysis, please consider to remove the description of RC8 as it would be confusing.

Lines 208-209. Please explain the author’s criteria here to remove RaTG13.

Fig. 2. Assign Fig. 2A, B, D.

Fig. 2. Provide amino acid and nucleotide numbers.

Lines 428-430. Generations of gene-engineered virus do not rely on particular strain. This claim here seems unworthy of your excellent work.

Reviewer #2: Dear PLOS ONE Editor

I accepted to review this article as I found this paper to be very interesting and worth to be published , however after including the structural mechanism to block Covid 19 infection which is the receptor binding domain (RBD) and the junction furin cleavage site( FCS) between S1 and S2, as stated in the paper. Mentioning the structural features required to block infection at these positions will strengthen the paper.

I suggest at the end of Results and Discussion paper to add a sub section paragraph referring to the RBD/ ACE2 and FCS mechanisms which invoke infection and relevant drug therapies, like angiotensin receptor blockers (ARBs) that block infection to enhance the strength of paper.

In particular,

The authors should mention the infection mechanism and ARBs potency in blocking virus infection using RBD /ACE2 and FCS as targets based on anionic structural features, like carboxylates and tetrazolates, required for inhibition and described in following papers suggested to be cited. This could be then a much useful paper for the scientific community.

Specifically, ARBs and bisartans with their negative structural features- warheads as tetrazolates in angiotensin receptor blockers (ARBs)- can block targets like RBD:ACE2 containing arginines and FCS by complexing with positive arginines at arginine rich multi basic sites S1/S2 ( 680-686) and S2(810-816) of furin which catalyse the cleavage of spike protein initiating infection. ARBs and especially bisartans with two tetrazolates inactivate the cleavage of furin by complexing to arginines, thus preventing infection. This is a key point in the mechanism of infection/ inhibition and worth to be mentioned. This will increase credibility and value of the paper.

Relevant papers to be cited

1. Ridgway et al, Expert Opinion on Therapeutic Targets, 1-23, 2024, DOI( available)

2. Ridgway et al Computational and Structural Biotechnology Journal, 20, 2091, 2022

3. Ridgway et al, Viruses, 14, 1029, 2022

4. Ridgway et al, Computational and Structural Biotechnology Journal, 21, 4589, 2023

5. Ridgway et al, Viruses, 15, 309, 2023.

6. Swiderski et al , Biomolecules, 13, 787, 2023

7. Kelaidonis et al, International Journal of Molecular Sciences, 24, 8454, 2023

8. Moore et al , Molecules, 27, 4854,2022.

9. Moore et al, Biomolecules, 11, 979, 2021

Authors should also mention in Conclusion Section, in a few words, the structural role in the infection- protection process.

After addressing the above, I agree and recommend this nice research to be published as an important contribution to the field and science.

John Matsoukas

Professor of Chemistry

Honorary Professor

Reviewer #3: Editor, Dated: 25/06/2024

PLOSONE

Reference No. :- PONE-D-24-20443

Title:- Emergence of crucial evidence catalyzing the origin tracing of SARS-CoV2

Greetings,

From my observation, the bioinformatics-related research article titled, “Emergence of crucial evidence catalyzing the origin tracing of SARS-CoV-2” makes no sense and is devoid of any novelty. This article has been written in a very bizarre manner. I am not able to understand what they want to convey. To my understanding, I have observed the below mentioned flaws. These are as follows:-

1. The abstract of this manuscript comprises of many mistakes such as, BANAL-20-52 Belong to. Belong to what?

2. The three strains of the SARS-CoV2 namely; BANAL-20-52, Rp22DB159, and S18CXBatR24 have been already identified.

3. One can’t claim purely on the basis of bioinformatics that we reject the laboratory-based engineering of SARS-CoV2 harboring furin-cleavage site between S1 and S2 subunit of the SARS-CoV2 spike protein.

4. Introduction section of this article has been written poorly with no coherence in order to comprehend it effectively and objectively.

5. Sentence under the Line number 54 and 55 is totally incorrect in terms of calculating the percentage of nucleotide identity.

6. References have been inserted within the sentences and sound very bad.

7. You can’t write, “Unfortunately, our research proposal was not funded”.

8. Do you think that readers have that much time to go through the Figure No. 3?

9. One can’t write materials and methods in result section.

10. In sentence number 423-424, you write unfortunately the genetic and geographical origin of SAR-COV2 remains unknown, resulting in suspicion about its origin. This sentence is contrary to your statement in the abstract section.

Overall, I am not happy with the findings, result section, and figures. I personally cannot recommend this paper for its possible consideration in PLOSONE journal.

Wishing you all the best

6. PLOS authors have the option to publish the peer review history of their article (what does this mean?). If published, this will include your full peer review and any attached files.

Reviewer #1: No

Reviewer #2: No

Reviewer #3: **Yes: **Dr Arif Bashir

Assistant Professor & Head

Department of Biotechnology & Clinical Biochemistry

Government College for Women, Nawa-Kadal

J&K-India

---

## [Author Response · Author response to Decision Letter 0]

15 Jul 2024

please see the Response to Reviewers.doc

---

## [Editor Report · Decision Letter 1]

15 Aug 2024

Emergence of crucial evidence catalyzing the origin tracing of SARS-CoV-2

PONE-D-24-20443R1

Dear Dr. Gao,

We’re pleased to inform you that your manuscript has been judged scientifically suitable for publication and will be formally accepted for publication once it meets all outstanding technical requirements.

Kind regards,

AbdulAzeez Adeyemi Anjorin, Ph.D.

Academic Editor

PLOS ONE
---

## [Editor Report · Acceptance letter]

21 Aug 2024

PONE-D-24-20443R1 

PLOS ONE

Dear Dr. Gao, 

I'm pleased to inform you that your manuscript has been deemed suitable for publication in PLOS ONE. Congratulations! Your manuscript is now being handed over to our production team.

Kind regards, 

on behalf of

Dr. AbdulAzeez Adeyemi Anjorin 

Academic Editor

PLOS ONE